# Advancing Radiobiology: Investigating the Effects of Photon, Proton, and Carbon-Ion Irradiation on PANC-1 Cells in 2D and 3D Tumor Models

**DOI:** 10.3390/curroncol32010049

**Published:** 2025-01-18

**Authors:** Alexandra Charalampopoulou, Amelia Barcellini, Giuseppe Magro, Anna Bellini, Sara Sevan Borgna, Giorgia Fulgini, Giovanni Battista Ivaldi, Alessio Mereghetti, Ester Orlandi, Marco Giuseppe Pullia, Simone Savazzi, Paola Tabarelli De Fatis, Gaia Volpi, Angelica Facoetti

**Affiliations:** 1Radiobiology Unit, Research and Development Department, CNAO National Center for Oncological Hadrontherapy, 27100 Pavia, Italy; anna.bellini@cnao.it (A.B.); sara.borgna@cnao.it (S.S.B.); giorgia.fulgini@cnao.it (G.F.); gaia.volpi@cnao.it (G.V.); angelica.facoetti@cnao.it (A.F.); 2Hadron Academy PhD Course, School for Advanced Studies (IUSS), 27100 Pavia, Italy; 3Radiation Oncology Unit, Clinical Department, CNAO National Center for Oncological Hadrontherapy, 27100 Pavia, Italy; amelia.barcellini@cnao.it (A.B.); ester.orlandi@cnao.it (E.O.); 4Department of Internal Medicine and Therapeutics, University of Pavia, 27100 Pavia, Italy; 5Medical Physics Unit, Clinical Department, CNAO National Center for Oncological Hadrontherapy, 27100 Pavia, Italy; giuseppe.magro@cnao.it; 6Radiation Oncology Department, Clinical Scientific Institutes Maugeri IRCCS, 27100 Pavia, Italy; giovannibattista.ivaldi@icsmaugeri.it; 7Research and Development Department, CNAO National Center for Oncological Hadrontherapy, 27100 Pavia, Italy; alessio.mereghetti@cnao.it (A.M.); marco.pullia@cnao.it (M.G.P.); simone.savazzi@cnao.it (S.S.); 8Department of Clinical, Surgical, Diagnostic and Pediatric Sciences, University of Pavia, 27100 Pavia, Italy; 9Medical Physics Unit, Clinical Scientific Institutes Maugeri IRCCS, 27100 Pavia, Italy; paola.tabarelli@icsmaugeri.it

**Keywords:** 2D cell cultures, 3D cell culture models, conventional radiotherapy, hadrontherapy, pancreatic cancer, radiobiology, spheroids

## Abstract

**Introduction:** Pancreatic cancer (PC) is one of the most aggressive and lethal malignancies, calling for enhanced research. Pancreatic ductal adenocarcinoma (PDAC) represents 70–80% of all cases and is known for its resistance to conventional therapies. Carbon-ion radiotherapy (CIRT) has emerged as a promising approach due to its ability to deliver highly localized doses and unique radiobiological properties compared to X-rays. In vitro radiobiology has relied on two-dimensional (2D) cell culture models so far; however, these are not sufficient to replicate the complexity of the in vivo tumor architecture. Three-dimensional (3D) models become a paradigm shift, surpassing the constraints of traditional models by accurately re-creating morphological, histological, and genetic characteristics as well as the interaction of tumour cells with the microenvironment. **Materials and Methods:** This study investigates the survival of pancreatic cancer cells in both 2D and spheroids, a 3D model, following photon, proton, and carbon-ion irradiation by means of clonogenic, MTT, spheroid growth, and vitality assays. **Results:** Our results demonstrate that carbon ions are more efficient in reducing cancer cell survival compared to photons and protons. In 2D cultures, carbon-ion irradiation reduced cell survival to approximately 15%, compared to 45% with photons and 30% with protons. In the 3D culture model, spheroid growth was similarly inhibited by carbon-ion irradiation; however, the overall survival rates were higher across all irradiation modalities compared to the 2D cultures. Carbon ions consistently showed the highest efficacy in reducing cell viability in both models. **Conclusions:** Our research highlights the pivotal role of 3D models in unraveling the complexities of pancreatic cancer radiobiology, offering new avenues for designing more effective and precise treatment protocols.

## 1. Introduction

Pancreatic cancer (PC) is a highly aggressive and lethal malignancy with a steadily increasing incidence [1]. The overall five-year survival rate for pancreatic cancer patients is less than 10%, and it is slightly better at approximately 13% for those with regionally advanced disease [2,3], mainly due to the frequent late-stage diagnosis. A major obstacle to early detection, even in high-risk groups, is that the pancreas often appears morphologically normal in the pre-diagnostic phase. Moreover, the disease can progress quickly and asymptomatically from subclinical stages to widespread metastasis, reducing the effectiveness of screening [4].

Approximately 90–95% of pancreatic tumors originate from the exocrine cells, with pancreatic ductal adenocarcinoma (PDAC) being the most common and aggressive subtype, accounting for 70–80% of cases [2]. Up to 50% of the patients were diagnosed with a metastatic form, 30% with a locally advanced or borderline resectable (bRe) disease, and only 20% with resectable (Re) disease.

In addition, despite the currently available therapeutic approaches depending on surgical resectability and the stage at diagnosis, a combined multimodal approach, including chemoradiotherapy, appears to improve local control [5,6]. However, PC exhibits significant intrinsic resistance to photon-based radiotherapy (RT) due to a higher hypoxic level and a stronger immunosuppressive tumor microenvironment compared to most solid malignancies [7]. In recent years, carbon-ion radiotherapy (CIRT) has shown promise in overcoming these hallmarks. A possible radiobiological explanation might be found in the higher relative biological effectiveness (RBE) of CIRT that allows for increased efficacy in cell-killing compared to photon beam RT, especially for highly hypoxic and radioresistant tumors, such as PDAC, also by the inhibition of angiogenesis [7,8]. On the other hand, from a clinical point of view, the dosimetric advantages of CIRT allow to improve the total dose to the target significantly and, at the same time, spare the high radiosensitive organs at risk (such as the duodenum and small bowel) [9]. Recently, the literature has highlighted that CIRT also plays a significant pro-immunogenic role by acting on the tumor microenvironment [10]. The scant prognosis and the encouraging data on CIRT call for the use of precise and real-time prediction models to direct tailored treatment plans. In the realm of radiobiology research, the two-dimensional (2D) cell culture models have a widespread implementation but fall short of replicating the complex three-dimensional (3D) architecture of tumors. In contrast, 3D models provide a more appropriate microenvironment for optimal cell proliferation, differentiation, and function, which are crucial for accurately understanding tumor biology and the response to RT [11,12]. Recent advancements in 3D cell culture technologies have opened new avenues for radiobiology research, particularly in pancreatic cancer. These 3D models, including spheroids, organoids, and biological or synthetic scaffolds, better mimic the in vivo tumor environment by maintaining critical features such as cell–cell and cell–matrix interactions, hypoxia, and nutrient gradients. These factors significantly influence tumor behavior and, importantly, the response to RT [13,14]. Given the lack of 3D in vitro studies on pancreatic cancer and the need for a more detailed understanding of the effects of RT on this challenging-to-treat malignancy, we evaluated the potential implementation of a 3D spheroid-based model in the radiobiological research on PDAC, starting with an initial 2D analysis. This study aims to assess the feasibility of using spheroids as a radiobiological model for pancreatic cancer by evaluating the responses to photons, protons, and carbon ions. Our research focuses on analyzing the differences in radiobiological responses between spheroids and traditional 2D models.

## 2. Materials and Methods

### 2.1. Cell Culture

The PANC-1 pancreatic carcinoma cell line, obtained from the Experimental Zooprophylactic Institute of Lombardy and Emilia Romagna, was cultured in a humidified incubator at 37 °C with 5% CO_2_. Cells were grown in Dulbecco’s Modified Eagle Medium (DMEM) supplemented with heat-inactivated fetal bovine serum, 100 U/mL of penicillin, and 0.1 mg/mL of streptomycin, with all reagents provided by Sigma Aldrich (St. Louis, MO, USA). Upon reaching confluence, the cells were detached using 10% trypsin for routine passaging.

### 2.2. Irradiation

Photon irradiation was performed using a 6 MV linear accelerator (LINAC) at the Radiation Oncology Department of Istituti Clinici Scientifici Maugeri (Pavia, Italy) at a dose rate of 3 Gy/min. Cells, seeded in T25 flasks filled with a non-complete medium, were irradiated at room temperature with doses of 0.5 Gy, 1 Gy, 2 Gy, 3 Gy, 4 Gy, 5 Gy, and 6 Gy. Flasks were placed horizontally on a 1.5 cm plexiglas platform in the homogeneity region of a 20 × 20 cm^2^ irradiation field and covered with a 1 cm thick water-equivalent bolus to maintain electronic equilibrium. The photon beam was directed from below at a 180° angle.

The proton and carbon-ion irradiation experiments were conducted at the National Center for Oncological Hadrontherapy (CNAO) in Pavia, Italy. Cells were irradiated in T12.5 flasks using a fixed horizontal clinical beamline and an active scanning dose delivery technique. A spread-out Bragg peak (SOBP) was optimized to achieve a homogenous (±2.5%) physical dose level across the target region by modulating 16 (130.6–163.6 MeV/u) or 31 (246–312 MeV/n) energies, with proton and carbon-ion beams, respectively. The T12.5 flasks were positioned at the center of the 6 cm modulated region of the SOBP inside a water phantom, with the entrance window aligned at the room’s isocentre. Cells were placed at a depth of 150 mm, corresponding to the middle of the SOBP [15]. Similarly, spheroids were cultured in 96-well plates and irradiated under comparable setup and beam configurations, using a solid water-equivalent phantom for practical purposes. An appropriate number of plastic slabs was used to cover the multi-well plates, providing a water-equivalent thickness of 150 mm.

### 2.3. Clonogenic Survival Analysis

After exposure to each irradiation type, the non-complete medium was replaced with a complete medium and the cells were incubated until further processing. Afterward, the cells were rinsed with PBS, trypsinized, and counted before being seeded at an appropriate density for each dose and incubated for 10 days to allow colony formation. After the incubation period, the medium was discarded, colonies were fixed with 70% fridge-cold ethanol, stained with 0.1% crystal violet solution (Sigma Aldrich), and counted. Colonies containing more than 50 cells were considered viable and were counted.

The surviving fractions (SF) were determined by dividing the number of counted colonies by the product of the plating efficiency (PE) and the initial number of cells seeded. These SF values were plotted on a semi-logarithmic graph as a function of the radiation dose. Clonogenic assay data were analyzed using the linear-quadratic (LQ) model to derive the parameters α, β, and the α/β ratio. For each irradiation type, the assay was conducted in triplicates across three independent experiments, ensuring the robustness and reproducibility of the data.

Clonogenic survival curves were generated using the Multiple Plot Visualizer v2.4 (14 March 2022) based on cell-plating efficiency and survival fractions. The cell survival data were fitted to the linear-quadratic (LQ) model: S = exp(−αD − βD^2^), where S represents the survival fraction, D is the dose in Gray, α (Gy⁻^1^) is the single-hit inactivation coefficient, and β (Gy^−2^) is the maximal double-hit inactivation coefficient. RBEs were obtained from the D10 values for the photons divided by the D10 for each particle.

### 2.4. MTT Assay

The effect of the different types of irradiation on PANC-1 cell lines’ viability was assessed by the 3-(4,5-dimethylthaiazol-2-yl)-2, 5-diphenyltetrazolium bromide (MTT) assay (Sigma Aldrich). Mock-treated (non-irradiated) and irradiated cells were seeded at a density of 3 × 10^4^ cells/well in 96-well plates containing a final volume of 100 μL/well at 37 °C in a 5% CO_2_ humidified incubator. At time points of 24 h, 96 h, and 5 days, 10 μL of the MTT solution were added to each well and then incubated at 37 °C in the dark for 4 h. The formazan crystals were dissolved in 100 μL of Dimethyl Sulfoxide (DMSO). The optical density (OD) was read at 570 nm by a microplate reader (BioTek^®^ 800TM TS). Data were presented as the percentage of cell viability as described below:

Percentage of cell viability = Average [OD570 treated cells]/Average [OD570 control cells] × 100 [16].

Each condition was assessed in triplicate wells, and the entire experiment was repeated three independent times to ensure consistency and account for variability. Data were reported as the percentage of cell viability relative to the mock-treated controls.

### 2.5. Spheroid Formation and Growth Evaluation

Spheroid formation was achieved by using U-bottom 96-well culture plates (Corning, Costar, NY, USA) and medium supplemented with 20% methylcellulose solution (Sigma Aldrich). Cells were seeded and cultured in a humidified atmosphere at 37 °C containing 5% CO_2_ for 4 days in order to form aggregates in a total volume of 100 μL, enough to fill each well. Afterward, spheroids were treated with single doses of different types of radiotherapy (RT). The medium was refreshed every 3 days, while spheroid growth was measured by means of the diameter length before and every 24 h after exposure to irradiation up to 4 days using the measurement tool of a phase-contrast microscope (Olympus IX71). For each experiment, a minimum of eight representative spheroids per irradiation dose were selected for each cell line, with the experiment being conducted three times to ensure reliability [17]. All values were normalized to the ones of the untreated samples [18].

### 2.6. Spheroid Viability

To monitor spheroid viability after the exposure to irradiation over time, we used the PrestoBlue cell viability reagent (Thermo Fisher Scientific, Waltham, MA, USA), which detects the reducing power of live cells 5 days post-irradiation. More specifically, the reagent was diluted 1:10 into the medium of each well of 96-well plates, which were then incubated at 37 °C and 5% CO_2_ for 10 min before reading on the absorbance microplate reader at 570 nm, using 600 nm as a reference wavelength, and the absorbance values were then normalized.

For each experiment, at least eight spheroids per irradiation type and dose were chosen, and each experiment was conducted three times [17].

### 2.7. Statistical Analysis

For our experiments, statistical analysis was performed using a two-way ANOVA followed by Tukey’s multiple comparisons test to evaluate the differences among groups or a two-tailed Student’s *t*-test to assess the significant differences between the two groups. Statistical significance was set at *p* ≤ 0.05. All statistical tests were performed using R 4.2.2.

## 3. Results

### 3.1. Clonogenic Survival

The PE and SF of PANC-1 cells were determined using the clonogenic assay, with the results assessed 10 days post-irradiation.

A dose-dependent reduction in SF was observed across all treatment modalities, including photons, protons, and carbon ions (Figure 1). We found a significantly lower SF of cells exposed to 4 Gy of proton (*p* = 0.0002) and carbon ions (*p* < 0.001) compared to those treated with photons, with the reduction being more pronounced for carbon ions.

More specifically, carbon ions demonstrated the highest RBE of 2, indicating their ability to induce cell death at lower doses compared to photons and protons. Protons exhibited an RBE of 1.3, reflecting similar effectiveness to photon irradiation, though still more effective than photons at comparable doses.

The lethal dose required to reduce the SF by 50% (LD50) for photon irradiation was estimated at approximately 4.3 Gy and 3.3 Gy for protons, while the LD50 for carbon ions was markedly lower at 2.1 Gy.

### 3.2. MTT Assay

The MTT assay results for the PANC-1 cells exposed to photons, protons, and carbon ions at doses of 2, 4, and 6 Gy were evaluated at different time points post-irradiation (Figure 2). At 24 h, no significant reduction in cell viability was observed for any radiation type, with viability remaining above 60% across all dose levels. By 96 h, a moderate decrease in cell viability was observed for all treatments, with carbon ions showing, at the higher doses of 4 Gy and 6 Gy, a more pronounced reduction compared to photons (66% for carbon ions compared to 70% for protons and 74% for photons at 4 Gy). After 5 days, a significant decrease in cell viability was observed, particularly for carbon ions, which demonstrated the most marked effect at all dose levels (i.e., 65% at 2 Gy compared to 85.4% of photons and 72% of protons). In contrast, compared to carbon ions, the viability rates were dose-dependent but significantly higher after the exposure of cells to protons (*p* = 0.014 for 2 Gy and *p* < 0.001 for 6 Gy).

### 3.3. Spheroid Growth

The effects of the three different types of irradiation on spheroid growth were also evaluated by measuring the diameter of the spheroids generated by the PANC-1 cell line (Figure 3). In the control group, the spheroid diameter increased steadily over time, reaching approximately 130% of the initial size by the 96 h time point. On the other hand, irradiated spheroids showed a dose-dependent decrease in diameter across all radiation types, with higher doses consistently leading to more pronounced reductions in spheroid size (Figure 4). While Figure 3 presents representative examples of individual spheroids, the quantification graph, based on data from multiple spheroids, provides a more accurate and comprehensive depiction of the overall effects.

More specifically, PANC-1 spheroids were less affected by photon or proton irradiation than carbon-ion irradiated ones, following the same trend of the 2D survival curves (Table 1). Carbon-ion irradiation demonstrated the most pronounced effect on spheroid growth, with the spheroids exposed to 6 Gy showing the greatest diameter reduction by 96 h (reduction to 51% compared to 57% for protons and 62% for photons). Photon and proton irradiation also caused a reduction in spheroid diameter, but the effect was less pronounced compared to carbon ions, particularly at the lower doses (a decrease of 81% for carbon ions in contrast to 85% for protons and 87% for photons). Notably, for photon-irradiated spheroids, the growth rates of the 2 Gy and 4 Gy groups did not differ significantly at any time point, indicating a similar response to these doses. However, for proton-irradiated spheroids, the 2 Gy dose showed a clear reduction of 21% in size by 72 h, with all doses displaying significant differences by 96 h (*p* = 0.0008 for 2 Gy and *p* < 0.0001 for 4 Gy and 6 Gy compared to the control samples).

### 3.4. Spheroid Viability

The viability of PANC-1 spheroids following irradiation with photons, protons, and carbon ions was also assessed (Figure 5). At all dose levels (2, 4, and 6 Gy), a dose-dependent decrease in spheroid viability was observed for all irradiation types. The photon and proton treatments showed a moderate reduction in viability (i.e., 0.85 and 0.80 at 4 Gy, respectively), with significant differences observed between 2 Gy and 4 Gy (*p* = 0.03). However, carbon-ion radiation induced a more pronounced effect (i.e., 0.61 at 4 Gy), with significant reductions between 2 Gy and 6 Gy (*p* = 0.03) and between 4 Gy and 6 Gy (*p* = 0.002) (Table 2).

## 4. Discussion

PDAC is currently a significant worldwide health challenge. It ranks seventh as the leading cause of cancer-related death and will overcome breast cancer as the third cause of cancer-related death by 2025 [19], necessitating enhanced research. Since PDAC is a malignancy that is intrinsically resistant to systemic and radiation treatments, research on alternative therapies is still ongoing. In the context of increasingly personalized medicine, and in light of promising data on local responses after CIRT for radioresistant histologies, we aim to understand PDAC’s response to different types of irradiation—photons, protons, and carbon ions—in order to improve the therapeutic options. Starting with a 2D model, we observed that clonogenic survival decreases in a dose-dependent manner when cells are exposed to each type of irradiation, but the impact of carbon ions is greater compared to protons and photons. Specifically, the lowest LD50 was recorded for carbon ions, and no advantage was observed for protons over photons in terms of the RBE. Carbon ions achieved an RBE of 2, effectively reducing cell survival to a greater extent than photons or protons across all dose levels. Our data fit with the literature evidence on the increased RBE of carbon ions for pancreatic cancer cell lines [20,21] and might partially explain the improved local control when pancreatic tumors, including locally advanced ones, are clinically treated with carbon ions rather than photons. Also, the significant reduction in viability found in our study after CIRT may support the promising clinical evidence.

However, although 2D models form the backbone of clinical research, they provide a limited perspective on the complex interactions within tumors. In comparison, 3D models, such as spheroids, offer a more realistic environment for studying radiobiological effects, enabling a deeper exploration of tumor behavior and treatment responses. [14,22,23]. These innovative models not only enhance our understanding of tumor biology but also offer a more predictive tool for evaluating radiotherapy effectiveness. Although the experiments performed with 3D spheroids confirmed that carbon ions cause a pronounced reduction in growth and viability, a trend consistent with the findings from the clonogenic and MTT assays in the monolayer cultures, PANC-1 in 3D models exhibited greater radioresistance. Indeed, at higher doses (6 Gy), we recorded a viability of 40% compared to the 33% seen in 2D. This is justified by the ability of the spheroid model to recreate a milieu closer to that found in vivo, where the immunosuppressive tumor microenvironment has a strong impact on the response to radiation treatment [24].

The tumor microenvironment in PDAC, which constitutes up to 80% of the total tumor mass, plays a pivotal role in fostering an immunosuppressive milieu that not only drives disease progression and metastasis but also contributes significantly to the chemo- and radioresistance of PDAC [25,26]. RT induces a broad spectrum of cellular and stromal effects beyond its cell-killing outcome [27]. One characteristic of cancer is the disruption of a highly structured tissue, resulting in localized necrosis and inflammation, gradients in interstitial pressure, and an uneven distribution of oxygen and nutrients [28]. RT induces a variety of alterations in tumor microenvironment networks, such as fluctuating hypoxia, immune system modulation, blood vessel regeneration, inflammation, and fibrosis [29]. Cancer cells have a selection advantage in this changed microenvironment, where they release immunosuppressive factors (i.e., Transforming growth factor β) and exhibit surface receptors (i.e., Programmed death ligand 1) that are able to inhibit T cells [30,31]. The greater radioresistance of cells in the spheroid can be justified by the creation of this unfavorable microenvironment [32]. The enhanced response after CIRT is also justifiable. In fact, it has been demonstrated that the pro-immunogenic factors released by RT are not sufficient on their own to trigger an immune-mediated anti-tumor response, especially when photon-based RT is used, whereas the response is stronger with carbon ions [10]. While in some conditions, the observed 5–6% differences in growth inhibition may appear modest, they hold significant biological relevance when considered within the context of microenvironmental complexity and inherent differential radiosensitivity of the cell line tested. However, spheroids composed of a single cell line limit the ability to effectively test these immunogenic properties. To overcome such a limitation, co-culture models incorporating stromal, endothelial, and immune cells could be used to better represent the intercellular interactions between different cell types [33]. Despite this limitation, using these higher-dimensional models for studying tumor-targeted therapies has yielded promising results [34]. The cellular interactions within spheroids lead to the formation of a 3D structure that closely mimics the native spatial organization and microenvironment of avascular tumors, where cells can proliferate, aggregate, and differentiate. Spheroids exhibit similar growth behaviors, cellular heterogeneity, and the formation of molecular gradients, among other characteristics. These distinctive features primarily result from their layered architecture, which consists of an inner necrotic core, a surrounding quiescent zone, and an outer layer of proliferating cells [35]. However, it is important to highlight there are common limitations in using cancer cell lines to generate spheroids. Cancer cell lines do not fully capture the diversity of individual patients, as each patient’s molecular profile is distinct. Additionally, cancer cell lines can accumulate genetic and epigenetic alterations over time, which may diverge from the characteristics of the original tumor cells, potentially influencing the outcomes observed in spheroid models [36]. This is particularly relevant when considering increased resistance to irradiation, as assessed by clonogenic survival, which has been proposed for cells cultured in 3D compared to 2D. This resistance is thought to result from enhanced repair mechanisms when cells are grown and treated in 3D environments. Nevertheless, it remains uncertain how this effect translates into the growth response of spheroids, where cells remain within the spheroid microenvironment after treatment, as opposed to being disaggregated and plated at clonal densities that do not accurately reflect the physiological conditions of tumors [37].

## 5. Conclusions

Our study confirmed that spheroids are a feasible environment for testing different radiation types, and the results obtained have led us to consider future analyses regarding studying the effects of the microenvironment on them. These data would be of great importance for evaluating tailored treatment in this challenging clinical context. Indeed, if, as demonstrated in other cell lines and in both in vitro and in vivo experiments [38,39,40], carbon ions prove to be superior in terms of efficacy to other forms of radiation, this could pave the way for combined treatments such as immunotherapy or chemotherapy. Integrating 3D models into pancreatic tumor radiobiology research represents a critical advancement beyond the limitations of traditional 2D systems [24], holding the potential to revolutionize treatment strategies for pancreatic cancer [41].

## Figures and Tables

**Figure 1 curroncol-32-00049-f001:**
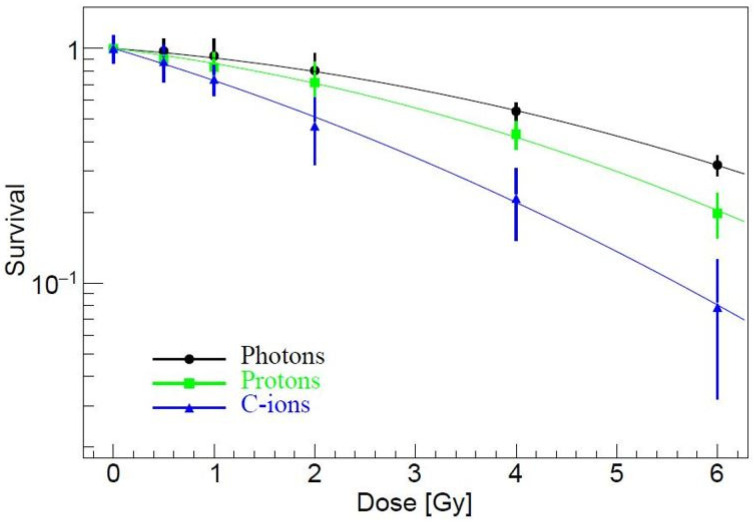
Survival curves of the clonogenic survival assays for PANC-1 cells after exposure to photons, protons, and C-ions. Error bars indicate ± standard deviations.

**Figure 2 curroncol-32-00049-f002:**
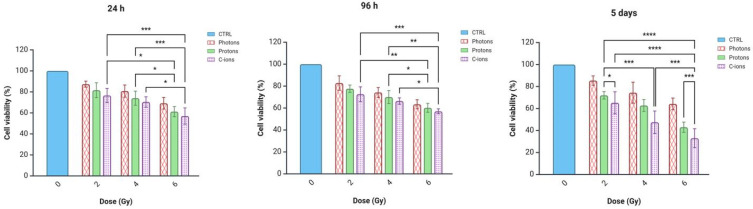
MTT assay results showing PANC-1 cell viability (%) at 24 h, 96 h, and 5 days post-irradiation with photons, protons, and C-ions at doses of 2, 4, and 6 Gy. Asterisks indicate statistically significant differences: *p* < 0.05 (*), *p* < 0.01 (**), *p* < 0.001 (***) and *p* < 0.0001 (****). Data are presented as mean ± standard deviation. Graphic created with Biorender.com (accessed on 28 November 2024).

**Figure 3 curroncol-32-00049-f003:**
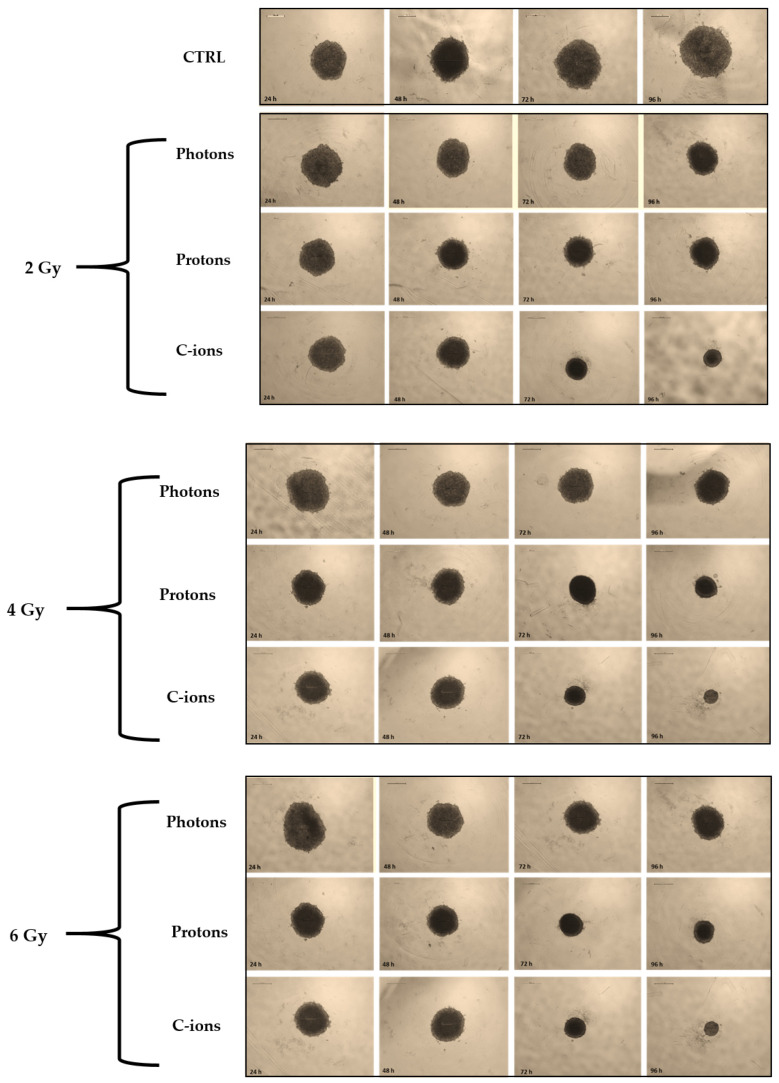
Representative images of PANC-1 spheroids under control condition (CTRL) and after exposure to 2 Gy, 4 Gy, and 6 Gy of photons, protons, and C-ions.

**Figure 4 curroncol-32-00049-f004:**
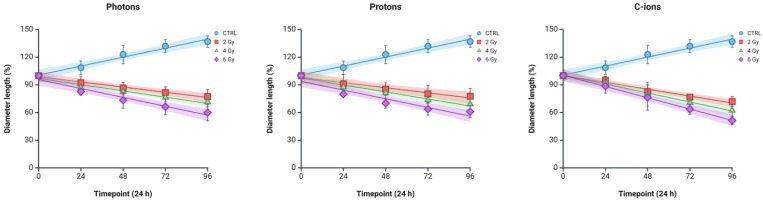
Spheroid growth following photon, proton, and C-ion irradiation. Baseline measurement (0) represents spheroid size before irradiation. Subsequent measurements were taken at 24 h intervals, tracking growth from 24 h to 96 h post-irradiation. Data are presented as mean ± standard deviation with best-fit lines and 95% confidence intervals. Graphic created with Biorender.com.

**Figure 5 curroncol-32-00049-f005:**
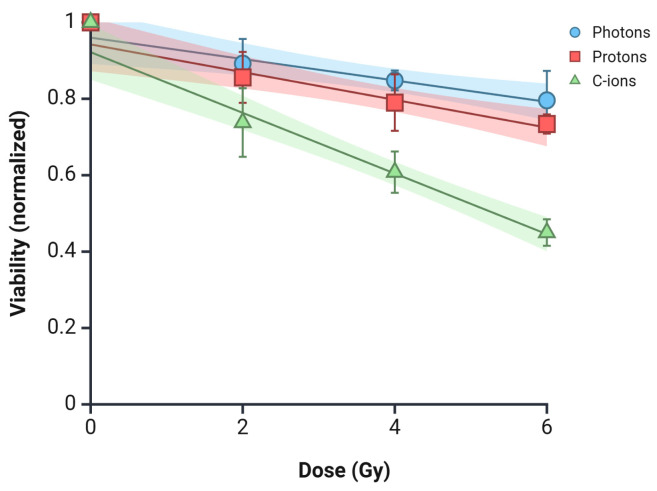
Spheroid viability 5 days post-irradiation with photons, protons, and C-ions. Data are presented as mean ± standard deviation with best-fit lines and 95% confidence intervals. Graphic created with Biorender.com.

**Table 1 curroncol-32-00049-t001:** Linear regression slopes of spheroid diameters for photons, protons, and carbon ions evaluated for the non-irradiated control group and the irradiated samples.

	**CTRL**	**2 Gy**	**4 Gy**	**6 Gy**	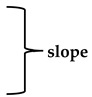
Photons	0.4056	−0.2313	−0.2833	−0.3931
Protons	−0.2333	−0.3167	−0.4014
C-ions	−0.3104	−0.4069	−0.5097

**Table 2 curroncol-32-00049-t002:** Linear regression slopes of spheroid viability for photons, protons, and carbon ions.

	Photons	Protons	C-Ions
Slope	−0.028	−0.03627	−0.07938

## Data Availability

The data supporting this study are available upon reasonable request from the corresponding author.

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
