# Peer review of "Advancing Radiobiology: Investigating the Effects of Photon, Proton, and Carbon-Ion Irradiation on PANC-1 Cells in 2D and 3D Tumor Models"

_curroncol, 2025, doi:10.3390/curroncol32010049_

Round 1

Reviewer 1 Report

Comments and Suggestions for Authors

Manuscript by Charalampopoulou et al is a useful resource of information on the effects of radiation on 2D and 3D cultures of PDAC cells in vitro. The manuscript sets up proposed doses and conditions that can be used as guidelines for future testing of combination therapies. However, the study uses only a single cell line, with no stromal components included, where the cell growth inhibition/death depend on the cell proliferation rate in 2D vs 3D conditions. With regard to the presented work, the title is too generalized, and should be changed to something that better reflects the data that is included and experimentally corroborated. For example: Effects of photon, proton and carbon-ion irradiation on PANC1 cell growth in 2D and 3D cultures in vitro.

In addition, several minor revisions are needed:

1. Please update the references 2, 3 with regards to PDAC patient survival

 For example  Siegel RL, Giaquinto AN, Jemal A. Cancer statistics, 2024. CA Cancer J Clin. 2024 Jan;74(1):12–49.

2. In the spheroid growth analysis in the photomicrographs  the effects of  C-ions seem dramatic compared to the photons and protons, while in the quantification graphs the differences do not seem to be that obvious. Please provide statistical analysis for the statement in line 217 PANC-1 spheroids were less affected by photon or proton irradiation than carbon ion irradiated ones, as based on the graphs in the Figure 4 that is not evident. Please show comparative graph for the three treatments at the same dose and rescale the graph to start from 0 for a better perspective of the inhibition of the spheroid size. Please comment whether the differences in growth inhibition that are 5-6% are biologically relevant.

3. Figure 5, please also rescale the axis to start form 0.

4. Please elaborate to a greater extent in the discussion that components of the PDAC TME make up the 80% of the PDAC tumor mass, and that the effects of radiation include tumor fibrosis, and list the disadvantages and limitations of use of just the cancer cells in spheroids as a model system for radiation in PDAC. 

Author Response

Comment 1: Manuscript by Charalampopoulou et al is a useful resource of information on the effects of radiation on 2D and 3D cultures of PDAC cells in vitro. The manuscript sets up proposed doses and conditions that can be used as guidelines for future testing of combination therapies. However, the study uses only a single cell line, with no stromal components included, where the cell growth inhibition/death depend on the cell proliferation rate in 2D vs 3D conditions. With regard to the presented work, the title is too generalized, and should be changed to something that better reflects the data that is included and experimentally corroborated. For example: Effects of photon, proton and carbon-ion irradiation on PANC1 cell growth in 2D and 3D cultures in vitro.

Response 1: Thank you for your thoughtful feedback on our manuscript. We acknowledge that the title "Advancing Radiobiology: Moving Beyond 2D by Integrating 3D Models in Pancreatic Tumour Research" may appear broad given the specific cell line and experimental conditions employed. However, the intent of the title was to underscore the paradigm shift from traditional 2D models to more physiologically relevant 3D systems in radiobiology research, particularly in pancreatic cancer.

That said, we understand the importance of ensuring that the title accurately reflects the content and scope of the study. Based on your suggestion, we are open to refining the title to better align with the experimental focus. A revised title could be: "Advancing Radiobiology: Investigating the Effects of Photon, Proton, and Carbon-Ion Irradiation on PANC-1 cells in 2D and 3D Tumour Models". This adjustment highlights the specific cell line and irradiation modalities while maintaining the emphasis on the integration of 3D models.

Regarding the use of a single cell line and the absence of stromal components, we fully agree that incorporating stromal elements would provide a more comprehensive understanding of the tumour microenvironment's role in radiobiology. While this is a limitation of the current study that we have clearly emphasized in the discussion, our aim was to establish a baseline framework using a well-characterized model system. Future studies will indeed focus on extending this work to include co-culture systems and additional cell lines to capture the complexity of the tumour microenvironment.

In addition, several minor revisions are needed:

Comment 2: Please update the references 2, 3 with regards to PDAC patient survival. For example Siegel RL, Giaquinto AN, Jemal A. Cancer statistics, 2024. CA Cancer J Clin. 2024 Jan;74(1):12–49.

Response 2: Thank you for pointing this out, we agree with this comment. Therefore, we have revised the references accordingly.

Comment 3: In the spheroid growth analysis in the photomicrographs  the effects of  C-ions seem dramatic compared to the photons and protons, while in the quantification graphs the differences do not seem to be that obvious. Please provide statistical analysis for the statement in line 217 PANC-1 spheroids were less affected by photon or proton irradiation than carbon ion irradiated ones, as based on the graphs in the Figure 4 that is not evident. Please show comparative graph for the three treatments at the same dose and rescale the graph to start from 0 for a better perspective of the inhibition of the spheroid size. Please comment whether the differences in growth inhibition that are 5-6% are biologically relevant.

Response 3: Thank you for your insightful comment and for highlighting the apparent discrepancy between the photomicrographs and the quantification graphs.

We would like to clarify that the photomicrographs depict a single representative spheroid from each experimental group, while the quantification graphs provide the average spheroid size across multiple samples. This distinction is crucial, as the photomicrographs are intended to illustrate morphology rather than serve as a basis for quantitative comparison.

Regarding the statement in line 217, we appreciate your observation about the apparent differences. To address this, we have clarified in the manuscript that the photomicrographs are representative examples and that the quantification graphs offer a more accurate and robust depiction of the treatment effects, as they account for variability across multiple spheroids.

In response to your suggestion, we have rescaled the quantification graphs to start from 0. However, we believe that adding a direct comparative graph for the three treatments at the same dose could lead to confusion, as it may oversimplify the complex dynamics observed. Instead, we have retained the slope analysis, which we believe provides a more nuanced understanding of the growth trends over time.

Finally, while the observed differences in growth inhibition (approximately 5-6%) may seem modest, we consider them biologically significant within the context of our study. This is particularly relevant given the differential radiosensitivity of the treatment groups. In the discussion section, we have commented on the relevance of these findings.

Comment 4: Figure 5, please also rescale the axis to start form 0.

Response 4: Thank you for your comment regarding the scaling of the graph. We understand the importance of consistency and the value of starting the scale at 0 for better interpretability. While our original intention was to highlight the slope and variations between the conditions using a different starting point, we have updated the graph per your suggestion to start the scale from 0.

Comment 5: Please elaborate to a greater extent in the discussion that components of the PDAC TME make up the 80% of the PDAC tumor mass, and that the effects of radiation include tumor fibrosis, and list the disadvantages and limitations of use of just the cancer cells in spheroids as a model system for radiation in PDAC.

Response 5: Thank you for your comment. We have revised the Discussion section accordingly.

Reviewer 2 Report

Comments and Suggestions for Authors

This manuscript addresses the challenging issue of pancreatic ductal adenocarcinoma (PDAC), emphasizing the potential of carbon ion radiotherapy (CIRT) and 3D culture models. The innovative focus on spheroid-based models for radiobiological studies aligns well with current research trends and has significant implications for the field. However, there are several areas that require clarification or improvement.

While statistical tests such as two-way ANOVA and Tukey's test are mentioned, the presentation of results lacks clarity. Specifically, exact p-values should be reported instead of general terms like "p < 0.05." Additionally, some comparisons do not include statistical significance values or confidence intervals, which would strengthen the reliability and interpretability of the findings.

Although the study aims to compare 2D and 3D models, it does not provide a clear quantitative or qualitative comparison of their responses to irradiation. The inclusion of specific metrics or visualizations, such as representative images of spheroids versus 2D cultures post-irradiation, would substantially enhance the discussion.

The manuscript demonstrates differential responses to various irradiation modalities but lacks a discussion on the underlying mechanisms. In particular, the observed differences in relative biological effectiveness (RBE) between photons, protons, and carbon ions remain unexplored. Incorporating molecular markers or pathway analyses could provide deeper insights and strengthen the scientific narrative.

The manuscript mentions the use of at least three independent experiments, but the variability between experiments and the number of replicates per condition are not clearly reported. Providing a detailed description of sample size calculation and variability would enhance the robustness and reproducibility of the findings.

Finally, pancreatic cancer is a lethal malignancy with poor prognosis and low survival rates. It would be beneficial for the authors to emphasize the critical need for early-stage diagnosis of pancreatic cancer. To support this discussion, the following reference could be cited: doi: 10.4251/wjgo.v16.i4.1256.

Author Response

Comment 1: This manuscript addresses the challenging issue of pancreatic ductal adenocarcinoma (PDAC), emphasizing the potential of carbon ion radiotherapy (CIRT) and 3D culture models. The innovative focus on spheroid-based models for radiobiological studies aligns well with current research trends and has significant implications for the field. However, there are several areas that require clarification or improvement.

While statistical tests such as two-way ANOVA and Tukey's test are mentioned, the presentation of results lacks clarity. Specifically, exact p-values should be reported instead of general terms like "p < 0.05." Additionally, some comparisons do not include statistical significance values or confidence intervals, which would strengthen the reliability and interpretability of the findings.

Response 1: Thank you for your comment. We have presented all the statistical significance values and corresponding confidence intervals obtained from our analysis, ensuring that the results are fully and transparently reported for clear and reliable interpretation. As suggested, we have revised the manuscript and added the exact p-values for all statistical comparisons, with the exception of cases where the p-value is less than 0.0001, as the software does not provide an exact value in these instances.

Comment 2: Although the study aims to compare 2D and 3D models, it does not provide a clear quantitative or qualitative comparison of their responses to irradiation. The inclusion of specific metrics or visualizations, such as representative images of spheroids versus 2D cultures post-irradiation, would substantially enhance the discussion.

Response 2: Thank you for your insightful comment regarding the comparison between 2D and 3D models in their responses to irradiation. We acknowledge the challenges in directly comparing metrics such as clonogenic survival in 2D cultures with the diameter and volume changes in 3D spheroids, as these represent fundamentally different endpoints that reflect distinct aspects of cellular and tissue-level responses to irradiation.

Nonetheless, our results provide clear evidence of the dose-dependent impact of irradiation across the models. In particular, the greater impact of carbon ion irradiation is consistently demonstrated, reflecting its higher linear energy transfer (LET) and the associated biological effectiveness. This important takeaway aligns with the study's objective to emphasize the differences in treatment efficacy across conditions using both 2D and 3D cell cultures.

Comment 3: The manuscript demonstrates differential responses to various irradiation modalities but lacks a discussion on the underlying mechanisms. In particular, the observed differences in relative biological effectiveness (RBE) between photons, protons, and carbon ions remain unexplored. Incorporating molecular markers or pathway analyses could provide deeper insights and strengthen the scientific narrative.

Response 3: Thank you for your insightful feedback. We appreciate your suggestion to explore the underlying mechanisms driving the differential responses to various irradiation modalities. While the current manuscript focuses primarily on demonstrating the observed effects on PANC-1 cells in 2D and 3D models, we agree that incorporating molecular markers or pathway analyses would provide valuable mechanistic insights. However, such analyses were beyond the scope of this study, which aimed to establish baseline data for comparative radiobiological effects in simplified systems.

That said, we acknowledge the importance of delving deeper into the mechanisms behind these effects. Future studies will investigate key molecular pathways and markers, such as DNA damage response (e.g., γH2AX), apoptosis (e.g., caspases), and hypoxia-related signaling, to elucidate the biological processes underlying the observed RBE differences.

Comment 4: The manuscript mentions the use of at least three independent experiments, but the variability between experiments and the number of replicates per condition are not clearly reported. Providing a detailed description of sample size calculation and variability would enhance the robustness and reproducibility of the findings.

Response 4: Thank you for highlighting the importance of reporting variability and sample size details. To address this, we have revised the Materials and Methods section.

Comment 5: Finally, pancreatic cancer is a lethal malignancy with poor prognosis and low survival rates. It would be beneficial for the authors to emphasize the critical need for early-stage diagnosis of pancreatic cancer. To support this discussion, the following reference could be cited: doi: 10.4251/wjgo.v16.i4.1256.

Response 5: Thank you for your comment. We have revised the manuscript accordingly.

Round 2

Reviewer 1 Report

Comments and Suggestions for Authors

Authors have sufficiently addressed the raised concerns.

Reviewer 2 Report

Comments and Suggestions for Authors

Congratulations to all the authors and thank you for addressing all comments and suggestions during the review process.